# A Portable, Negative-Pressure Actuated, Dynamically Tunable Microfluidic Droplet Generator

**DOI:** 10.3390/mi13111823

**Published:** 2022-10-25

**Authors:** Martin Trossbach, Marta de Lucas Sanz, Brinton Seashore-Ludlow, Haakan N. Joensson

**Affiliations:** 1KTH Royal Institute of Technology & Science for Life Laboratory, 17165 Solna, Sweden; 2Karolinska Institutet & Science for Life Laboratory, 17165 Solna, Sweden

**Keywords:** droplet microfluidics, plug & play, portable microfluidics, spheroids, microtissues

## Abstract

Droplet microfluidics utilize a monodisperse water-in-oil emulsion, with an expanding toolbox offering a wide variety of operations on a range of droplet sizes at high throughput. However, translation of these capabilities into applications for non-expert laboratories to fully harness the inherent potential of microscale manipulations is woefully trailing behind. One major obstacle is that droplet microfluidic setups often rely on custom fabricated devices, costly liquid actuators, and are not easily set up and operated by non-specialists. This impedes wider adoption of droplet technologies in, e.g., the life sciences. Here, we demonstrate an easy-to-use minimal droplet production setup with a small footprint, built exclusively from inexpensive commercially sourced parts, powered and controlled by a laptop. We characterize the components of the system and demonstrate production of droplets ranging in volume from 3 to 21 nL in a single microfluidic device. Furthermore, we describe the dynamic tuning of droplet composition. Finally, we demonstrate the production of droplet-templated cell spheroids from primary cells, where the mobility and simplicity of the setup enables its use within a biosafety cabinet. Taken together, we believe this minimal droplet setup is ideal to drive broad adoption of droplet microfluidics technology.

## 1. Introduction

Compartmentalization of cells or biomolecular samples into monodisperse reaction vessels via droplet microfluidics offers unique benefits to chemical and biological analysis and screening. Key among these benefits are improved throughput, enhanced sensitivity of detection due to the reduced dilution in these small reaction volumes, and the minimization of reagent and sample consumption [1].

Droplet microfluidics, a versatile and high-throughput platform technology, employs pico- to nanoliter-sized aqueous droplets in an immiscible fluorinated oil phase [2]. Fluorinated oils, such as Novec HFE-7500, have several advantageous properties, such as high gas solubility [3] and the availability of biocompatible surfactants [4], making them the preferred choice for biological assays. These biocompatible surfactants usually consist of a fluorophobic-hydrophilic polyethyleneoxide head group and fluorophilic-hydrophobic tails. These amphiphilic characteristics lead to them populating the interface and stabilizing the metastable droplet by lowering the surface tension, preventing coalescence [5]. The droplets are stable over extended incubation periods [6] and at higher temperatures [7,8].

A plethora of applications and assays have been developed using droplet microfluidics. Single cell assays [9,10], droplet PCR [11,12] and viral detection [13], among many others, demonstrate the versatility of the platform and the potential for customizable workflows. Entire microfluidic pipelines are created by combining several unit operations, e.g., combining droplet production, splitting, or sorting, in sequence. Aided by the control offered by microfluidics, a specifically optimized microfluidic device has been developed for virtually every individual unit operation. Moreover, there have been advances in chip design and flow rate management to enable an even wider range of applications [14,15].

Recently, microfluidic droplets have been explored as vehicles for tissue engineering, e.g., in spheroid assembly [16] or gel encapsulation applications [17,18], and are even gaining traction in research studying stem cell differentiation [19,20]. The droplets used in these applications are usually larger than droplets used for single cell studies to allow for encapsulation of sufficient cell numbers [6] or ensure adequate nutrient supply to maintain cell viability for multi-day droplet culture [21]. Addressing the need for scaling, we previously published an automated workflow for high-throughput spheroid production in microfluidic droplets using a liquid handling robot [6], and presented a production optimization pipeline employing a deep neural network to characterize spheroid morphology [22].

There are numerous active and passive methods for producing aqueous droplets in a continuous phase of which flow-focusing droplet generation is most widely used [23,24]. In negative-pressure droplet generation, a pressure source set below ambient pressure is applied at the device outlet to actuate droplet generation. While positive-pressure or positive flow-driven droplet generation using flow-focusing techniques have been extensively studied, including analysis of the impacts of geometry and flow regimes on droplet size [25,26,27,28,29], negative-pressure operation has received less attention. Conceivably, the general tenets still apply, but conditions such as hydraulic resistances become more impactful [30]. Crucially, while the flow rates of the dispersed and continuous phase are directly controllable in positive-pressure or flow-driven droplet generation, they are not independent input parameters for negative-pressure droplet generation.

However, negative-pressure droplet generation has some key advantages over the more conventional positive-pressure-driven methods. Actuated from the collection outlet, these methods allow for continuous access to the oil and sample inlets and for interfacing with liquid handling robots, enabling automation. Compared to syringe pump-driven droplet generation, these advantages are even more pronounced, and with the added advantage of drastically reduced setup and turnaround time. Hundreds of thousands of droplets, each a compartmentalized reaction vessel, can be produced in a matter of minutes while the samples can be agitated in the inlet reservoir, thus preventing particle aggregation, striation, and sedimentation. This is particularly advantageous for sensitive biological samples, such as mammalian cells.

Mammalian cell culture is dependent on sterile conditions and is often sensitive, as some cells, especially primary or stem cells, react negatively to prolonged exposure to lower-than-physiological temperatures [31,32,33]. In addition, some antibiotics are light-sensitive, which adds to the need for fast experimental processing for steps outside of the incubator [31]. It follows that any cell processing step should be rapid, and the instrument should be deployable in a constrained space, such as a biosafety cabinet, allowing for a fast turnaround time. Naturally, this not only applies for normal cell culture steps, such as freezing, thawing, and upkeep, but for microfluidic operations as well—possibly even more so, as the cells have to resist additional stresses, such as shear forces. Several commercial providers offer relatively compact pressure controller devices that are used extensively within the microfluidic community. However, they still require additional infrastructure, such as an external power supply and pressure source. This might not pose a challenge to research groups focusing on microfluidics, but it adds a layer of complexity and thus a hurdle to overcome for less expert laboratories wanting to adopt microfluidic techniques.

Thus, while a host of operations is available to a specialized lab with appropriate equipment and sufficient expertise—including, but not limited to, generation, incubation, injection, splitting and sorting—the lack of standardization and use of bulky, customized experimental setups limit the dissemination of these powerful techniques to a wider range of laboratories. This creates a disconnect between what is technically possible today and what is easily available and readily used—both limiting advances in research areas that could benefit from microfluidics tools and hampering awareness of biological users’ needs in the droplet microfluidics community.

In order to enable non-specialized labs to make use of droplet microfluidics, we compiled and thoroughly characterized a minimal droplet generation setup using only low-cost, commercially available equipment, with a low barrier to entry in mind. Actuation of liquids by exerting variable (negative) pressures at the outlet and inlets of the system enables wide-ranging tuning of droplet sizes and dynamic changes in droplet composition, using a single circuit geometry. The presented setup can produce a range of droplet rates, sizes, and compositions on demand, providing versatility while maintaining a minimal footprint. We further demonstrated the utility of the setup by encapsulating primary hepatocytes in droplets to generate cell spheroids for screening purposes. As shown, this setup addresses the needs of the wider biomedical research community and encourages the adoption of microfluidic workflows and furthers the democratization of healthcare by, e.g., decreasing costs for screening campaigns.

## 2. Materials and Methods

### 2.1. Droplet Generation Setups

Three different experimental setups were tested and configured as shown in Figure 1. Briefly, a single pump allows for limited size tuning by adjusting the applied pressure at the collection pump; an additional second pump attached to the continuous phase reservoir increases the accessible droplet size (Figure 1A).

The microfluidic device, the Fluidic 163 (microfluidic ChipShop), features a channel depth of 175 µm, and 140 µm-wide channels for the aqueous and oil phases, respectively. The channel widens to 420 µm 200 µm downstream of the junction (see also Appendix A). Attaching two pumps at the dispersed phase reservoirs allows for on-demand tuning of the droplet composition. All necessary microfluidic components are shown, and a detailed list of parts can be found in the Appendix A. The reservoir container lids connected to pumps were sealed using circular Parafilm cut-outs, prohibiting air leakage. The entire setup is portable (see Appendix A) with a compact footprint.

### 2.2. Pump Characterization

The mp-gas+ membrane pump was connected to a Pressure Unit S (Fluigent) via microfluidic PEEK tubing (Zeus) and the pressure was recorded using OxyGEN software (Fluigent). The frequency response was assessed in 10 Hz increments between 50 and 800 Hz; the amplitude response was assessed in 2 AU increments between 18 and 250 AU.

### 2.3. Droplet Rate Measurements

Fluorinated oil (Novec HFE-7500, 3M) was supplemented with 2% *w*/*v* 008 surfactant (Ran Biotechnologies) for the continuous phase. Fluorescein (Sigma) at a concentration of 20 µM in phosphate buffered saline (PBS, Medicago) was used as the aqueous phase.

The microfluidic chip was connected to tubing as shown in Figure 1A. For the single-pump setup, the mp-gas+ (Bartels Mikrotechnik GmbH) pump at the collection outlet was set to 4.61–13.78 kPa of negative pressure. For the dual-pump setup, the collection pump was set to 13.78 kPa of negative pressure and the mp-gas+ pump attached to the oil inlets was set to 0.3–1.4 kPa of negative pressure.

We focused a 514 nm laser just after the widening of the channel downstream of the nozzle, collected the fluorescence emission using a photomultiplier tube (Hamamatsu), and recorded the fluorescent signal using a custom LabView program (National Instruments).

### 2.4. Droplet Size Measurements

The microfluidic setup was employed in the same way as described in Section 2.3. To transfer the generated emulsion afterwards, a syringe attached at the Luer T connector is manually aspirated, emptying the collection tank. The emulsion was then imaged using a Ti-E Eclipse (Nikon) and droplet sizes were manually measured using ImageJ [34].

### 2.5. Flow Rate Measurements

The microfluidic setup was employed in the same way as described in Section 2.3. The pumps were started and the emulsion collected. After 2 min, the pumps were stopped and disconnected to remove any pressure differential. The collected emulsion was broken using an antistatic gun, Zerostat 3 (Milty), according to published protocols [35], and the phases were separated into different tubes. The mass of the samples was measured using a high-precision scale (Ohaus) and their volume was calculated considering the densities of both phases (aqueous phase: 1000 kg/m^3^, continuous phase: 1614 kg/m^3^).

### 2.6. Numerical Simulation of Negative Pressure Droplet Generation

To numerically study droplet generation using flow-focusing and negative pressure applied solely at the collection outlet, a 2D approximation of the nozzle region was implemented in COMSOL Multiphysics (COMSOL AB), using the respective densities and dynamic viscosities for water and Novec HFE-7500. Virtual channel boundaries were made using acrylic plastic, and the contact angle was set to superhydrophobic 0 rad to exclude any wetting behavior. The simulation encompassed the first 1.25 s after application of a pressure differential in 5 ms increments. The simulation was carried out on a 2020 M1 16 GB RAM Macbook Pro (Apple).

### 2.7. Droplet Content Manipulation

We used the same continuous phase as described in Section 2.3. Fluorescein (Sigma) at a concentration of 20 µM in PBS (Medicago) was used as aqueous phase 1 and 15 µm Fluoro-Max Red Dry Fluorescent Particles (ThermoFisher Scientific) dispersed in Milli-Q water were used as aqueous phase 2.

The microfluidic chip was connected to tubing as shown in Figure 1B. The mp-liq pumps (Bartels Mikrotechnik GmbH) connected to the aqueous phase reservoirs were dynamically controlled, we regulated the amplitude parameters of one pump from 24 AU up to 118 AU, while regulating the other from 118 AU down to 24 AU in a corresponding, opposed fashion. A mp-gas+ pump (Bartels Mikrotechnik GmbH) was connected at the collection outlet and set to −13.8 kPa. The emulsion was imaged using a Ti-E Eclipse (Nikon) and ImageJ [34]. Median droplet fluorescence and particle number were manually measured using ImageJ.

### 2.8. Cell Spheroid Production

A vial of frozen primary hepatocytes (Lonza) was thawed according to the manufacturer’s instructions and resuspended in Williams E media with added penicillin-streptomycin (100 U/mL), L-Glutamine (200 mM), insulin (10 µg/mL), 0.1 µM dexamethasone, 5.5 µg/mL transferrin, and 10.95 ng/mL sodium selenite, at a concentration of 7 × 10^6^ cells/mL. An amount of 400 µL of cell-containing media was encapsulated in Novec HFE-7500 (3M), supplemented with 2% *w*/*v* 008 surfactant (RAN Biotechnologies) using a variation of the single-pump configuration (see Appendix A) operated at a frequency of 560 Hz and an amplitude of 250 AU, including a 3 mm magnetic stir bar in the cell loading chamber, and we positioned the aqueous reservoir on a rotating magnetic actuator inside a laminar air flow cabinet. Following droplet generation, the emulsion was transferred to a 5 mL syringe and incubated upright in a cell culture incubator with 5% CO_2_ and 100% humidity, with a Minisart 5 µm filter (Sartorius) attached to the syringe to allow for gas exchange. Droplet samples were retrieved from the incubation syringe immediately following droplet transfer and after 48 h of incubation. Following 48 h of incubation, spheroids were recovered from the emulsion by filtering on a PTFE membrane (Sartorius) according to a previously published protocol [6]. Spheroids were counted, resuspended in fresh media, and deposited as 20 spheroids per well into a 384 well plate.

### 2.9. Viability Measurements

A CellTiterGlo (Promega) assay was performed to assess viability. A total of 30 µL of CellTiterGlo was added to 40 µL of standards or spheroids in media and incubated for 1 h at 37 °C and read on an EnSight Multimodal plate reader (Perkin Elmer).

### 2.10. Albumin Secretion

Albumin levels were measured to quantify albumin secretion. A total of 40 µL of supernatant was frozen at −20 °C until analysis using an Albumin (human) LANCE Ultra TR-FRET Detection Kit on an EnSight plate reader (PerkinElmer) using the manufacturer’s protocol.

## 3. Results

### 3.1. Pump Characterization

To characterize micropumps that provide actuation in our negative pressure setup, we measured the pressure differential response to the applied membrane frequency and amplitude across the available ranges of frequencies (50–800 Hz) and amplitudes (0–250 AU). When actuated with an amplitude of 150 AU and 250 AU, the micropumps could deliver pressure differentials in the range from −2.8 kPa to −13.7 kPa, with the highest available pressure differential observable occurring with the maximal amplitude of 250 AU and in the frequency range 525–575 Hz (Figure 2A). Notably, the lowest accessible frequencies demonstrated a higher degree of variation in their pressure output. For that reason, and due to considerations with regards to the maximally attainable differential pressure, we chose to examine the frequencies of 560 Hz and 800 Hz more closely.

The pressure differential readings from the micropumps demonstrate a stepwise response with a step width of 8 AU for amplitude modulation. Measurable negative pressures were recorded at amplitudes of 17 AU and above for all tested pumps. At 800 Hz, the pressure differential increases at the top end of the amplitude setting were more incremental than at 560 Hz (Figure 2B). We speculate that this might be caused by the high wave frequency, which might limit the travel of the membrane generating the pressure, thereby decreasing the effective amplitude. Based on these findings, we chose to use 560 Hz as the set frequency for the remainder of the experiments and use the amplitude parameter to control pump output. With a set micropump actuation frequency, pressure outputs can be modulated with a pressure step size of 0.5 kPa.

### 3.2. Characterization of Droplet Generation

#### 3.2.1. Single-Pump Operation

In order to gain conceptual insight into the droplet generation behavior of negative pressure actuation applied at the collection outlet with open inlet reservoirs (cf. Figure 1A), we performed numerical simulation using COMSOL Multiphysics. In order to enable simulation within a reasonable amount of time and computational resource use, we limited the simulation to a two-dimensional approximation of the nozzle region. We simulated real-world behavior by setting a surface tension coefficient of 2 mN/m and densities of 1000 kg/m^3^ and 1614 kg/m^3^, as well as dynamic viscosities of 1 mPa·s and 1.243 mPa·s for the water and fluorinated oil phases, respectively. To avoid wetting phenomena, the contact angle was set to 0 rad.

The results show a trend of decreasing droplet size and increasing droplet frequency with increasing pressure differential applied at the outlet. In other words, the volume of the droplets decreased as the rate increased. Notably, the rate increased faster than the droplet volume decreased, leading to an increased flow rate for the aqueous phase. The oil flow rate was less affected by the higher-pressure differential, resulting in an overall increase in the water-to-oil ratio (see Figure 3).

Using the single-pump configuration, we set out to validate the observations gained from the numerical simulation with the microfluidic setup and noted the same trend: when generating an emulsion using only a single pump attached at the collection outlet, the rate of droplet generation correlated with increasing pressure differential, whereas droplet volume was inversely correlated. Notably, this behavior is consistent with positive pressure generation in the dripping regime, where an increase in flow rates leads to smaller droplets [36]. The oil flow increased as well, but not to the same extent as the aqueous phase flow, leading to an increase in the aqueous-to-oil volume ratio from 0.1 at −4.6 kPa to 0.6 at −13.8 kPa. In this configuration, we were able to generate droplets of 3.1 nL to 5.3 nL at rates of 560 Hz to 114 Hz, respectively (Figure 4A,B). Operating the setup from a dead stop for 2 min, we weighed the phases separately (Figure 4C).

We observed less aqueous phase than expected from droplet rate and droplet volume measurements (for −4.6 kPa, 36 µL expected versus 15 µL measured; for −13.8 kPa, 103 µL expected versus 93 µL measured), which we attributed to the initial equilibration of flows. Due to the higher density of the fluorinated oil, the aqueous phase is pushed back into the reservoir when the system is not pressurized. Only after a negative pressure differential is applied at the outlet, the aqueous phase (re-)enters the chip and moves downstream. This takes longer for lower pressure differentials, accounting for the larger discrepancy of 59% when compared to the 9% discrepancy of the −13.8 kPa sample.

From the channel dimensions of the microfluidic chip, we were able to approximate the relative resistances of the channels when operated as a droplet generation device. Assuming the channel height to be 175 µm, as stated by the manufacturer, we calculated a combined resistance of ~32 kPa m^−3^ s^−1^ for the oil channels and ~30 kPa m^−3^ s^−1^ for the aqueous channels until the junction where both converge.

We assume both phases to be Newtonian, since no large quantities of polymers were dissolved in the aqueous phase. With regards to the fluorinated oil, HFE-7500 has minute non-Newtonian properties, but these should be negligible in our context due to the high shear forces necessary for this behavior to make a significant impact [37]. Consequently, a constant ratio between the two flows should follow if these were the only governing principles. However, we observed diverging behavior, much like in the numerical simulation. Confirming previous findings [22], we measured a greater increase in the flow of the aqueous phase compared to the continuous oil phase with increasing ∆*p* at the outlet. This means that while at an applied pressure difference of −4.6 kPa the ratio of water to oil is 0.11, this increases to 0.61 at −13.8 kPa, reproducing the trend observed in the 2D-simulation. We assume this to be a result of Laplace pressure at the nozzle interface: the effective pressure difference between the aqueous inlet and the water-oil interface ∆*p_a eff_* is decreased by the Laplace pressure necessary to balance the surface tension. This means that the derivative of ∆*p_a eff_* decreases faster than the derivative of the derivative of the pressure difference from the inlet to the interface at the nozzle ∆*p_o_*, or ∆*p_nz_*, with decreasing applied pressure at the outlet. As the flow rates *Q* scale with these derivatives, a change in the flow rate ratio is observed (Figure 5).

The Laplace pressure *p_lp_* is defined as
(1)$$p_{lp}=\frac{2\gamma }{r}$$
where *γ* denotes surface tension and r the radius of the interface.

The flow rates *Q_a_* and *Q_o_* for the aqueous and oil phases are dependent on both the pressure differential and the respective resistances:(2)$$Q_{a}=\frac{\left(\Delta p_{nz}-\Delta p_{lp}\right)}{R_{a}}$$
(3)$$Q_{o}=\frac{\left(\Delta p_{nz}-\Delta p_{Pump}\right)}{R_{o}}$$
where ∆*p_nz_* is the pressure difference between the inlet and the aqueous-oil interface at the nozzle, ∆*p_lp_* is the pressure difference over the interface, *R_a_* is the resistance of the aqueous channel from the inlet until the interface, *p_Pump_* is the pressure exerted by the pump, and *R_o_* is the resistance of one oil channel from the inlets up until the interface.

Utilizing Equations (2) and (3), we arrive at Equation (4), describing the ratio of flows:
(4)$$\frac{Q_{a}}{Q_{o}}=\frac{\left(\Delta p_{nz}-\Delta p_{lp}\right)}{R_{a}}\cdot \frac{R_{o}}{\Delta p_{nz}}=\frac{R_{o}}{R_{a}}\cdot \left(1-\frac{\Delta p_{lp}}{\Delta p_{nz}}\right)$$

Considering that the hydrodynamic resistances of the microfluidic channels do not change, and that both inlets are open and thus subject to atmospheric pressure, and approximating the Laplace pressure to be constant, we can thus conclude that the flow ratio of the aqueous to continuous phase increases with increasing ∆*p_nz_*, which is a direct result of an increase in ∆*p_e_* due to a decrease in *p_Pump_*.

Droplet generation using this single-pump setup ceased with pressure differentials lower than approximately −3 kPa, presumably because of the same Laplace pressure. Therefore, to limit excessive oil consumption, increase aqueous phase throughput, and access larger droplets for, e.g., spheroid production, a different strategy must be pursued.

#### 3.2.2. Dual-Pump Setup for Size Tuning

With a counteracting pump connected to the oil reservoir supplying the oil inlets (cf. Figure 1A), which thereby lowers the effective pressure differential, the oil flow can be modulated. This, in turn, affects the resulting droplet size. Running the collection pump at an amplitude of 250 AU to generate −13.8 kPa, we varied the pressure exerted from the pump connected to the oil reservoir from −0.3 to a maximum of −1.4 kPa, beyond which flow through the oil channels ceased and only aqueous phase was collected. This way, we were able to increase the droplet volume up to 21 nL, or to a diameter of 340 µm. Notably, the flow rate for the aqueous phase remained near constant at 90 µL min^−1^, while the oil flow rate steadily decreased from 143 µL min^−1^ to 43 µL min^−1^. In this configuration it appears that the rate of increase in droplet volume is linearly correlated to the decrease in droplet rate, resulting in the observed behavior of near-constant aqueous flow (Figure 6). This is in accordance with previous findings that droplet size in devices with a flow-focusing orifice is governed by geometry and the flow of the continuous phase [29,38].

#### 3.2.3. Three-Pump Setup for Droplet Composition Manipulation

To further study the versatility of the setup, we implemented a configuration with one collection pump and a counteracting pump at each aqueous inlet (cf. Figure 1B). We used a suspension of 1.5 × 10^6^ particles as a model for a concentrated mammalian cell culture sample and fluorescein dissolved in PBS as the second aqueous component to be encapsulated. Above amplitude 118 we observed flow from the junction towards the pump-actuated aqueous inlet; thus, we set this as the upper boundary and sequentially adjusted the pump settings from amplitudes 118/0 to 0/118 (see Figure 7A). As a result, we were able to rapidly produce an emulsion with a variety of droplet compositions without changing the sample (see Figure 7B,C).

#### 3.2.4. Single-Pump Setup for Microtissue Production in Biosafety Cabinet

Finally, we intended to verify the practicality of the minimal droplet generator setup for use with a biological sample in the sterile conditions necessary for mammalian cell culture work. Crucially, the minimal droplet generator can be operated by non-experts with very little training, has a minimal footprint, and can be quickly set up. Operating the pump at a frequency of 560 Hz and amplitude of 250 AU (for configuration see Appendix A), we encapsulated primary human hepatocytes and successfully demonstrated spheroid assembly after 48 h of droplet incubation (Figure 8A–C). High viability is maintained throughout the incubation period, and subsequent transfer and 120 h incubation in a GrowDex format suggests proliferation (increased ATP) (Figure 8D).

Albumin secretion is a hallmark of primary human hepatocyte function and is not retained in all culture models [39]. However, when compared to spheroid formation in an ultra-low attachment (ULA) plate format, microfluidic droplet-derived spheroids exhibit increased albumin secretion as assessed through an albumin quantification assay (Figure 8E). Taken together, these results suggest the microfluidic droplet-derived spheroids are suitable for downstream usage in, e.g., drug or toxicity screens.

## 4. Conclusions

Here, we characterized and applied a low-cost, minimal footprint droplet generation setup using one to three small membrane pumps. The modular setup is capable of producing monodisperse droplets for a wide range of droplet sizes (see Appendix A) and can be dynamically adjusted to vary droplet composition. All parts used are readily available and replaceable. The setup does not require a dedicated microfluidic laboratory infrastructure and can be set up virtually anywhere. All components that come into contact with the sample can be sterilized and the setup as a whole is sufficiently compact to be used in a sterile environment, such as a biosafety cabinet. We demonstrated this by installing the setup in a BSL2 cell culture hood and encapsulating particularly sensitive primary cells for droplet-assisted spheroid formation.

This minimal droplet generator setup represents a step forward in providing access to high-end droplet microfluidics to non-expert users, particularly in the life sciences, with a minimal barrier to entry. The deployment of inexpensive and accessible microfluidics in life science laboratories should yield novel user perspectives to spawn new development and novel technology use cases.

## Figures and Tables

**Figure 1 micromachines-13-01823-f001:**
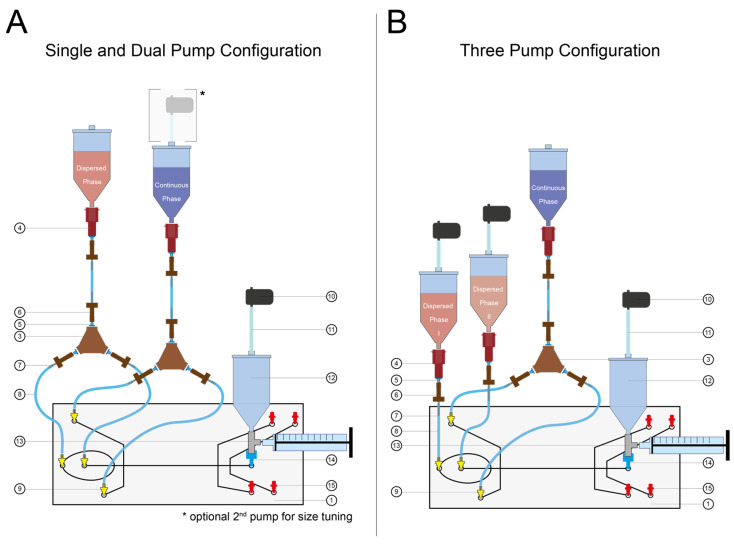
Schematic depictions of the minimal droplet generator setup. (**A**) Illustration of single- and dual-pump configuration. The depiction shows how the components are assembled for operating with one or two pumps; when a second pump is added to the continuous phase reservoir, larger droplets can be generated. (**B**) Schematic depiction of three-pump configuration. The illustration shows how the components are assembled for operating with three pumps. The numbers refer to the part list (see Appendix A). Tubing length is customizable, however, we used lengths of 7) 3 cm and 8) 5 cm, with enough overlap to ensure seal. The collection syringe can be any syringe with a Luer lock.

**Figure 2 micromachines-13-01823-f002:**
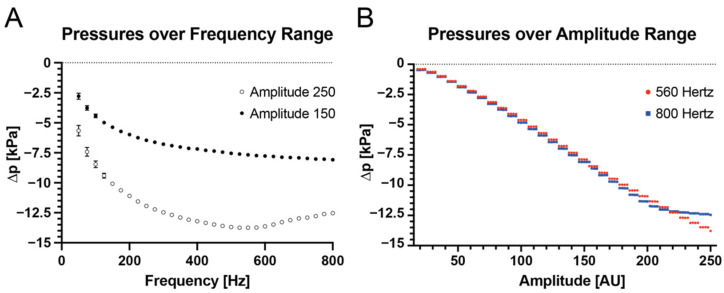
Pump characterization. (**A**) Micropump pressure outputs at fixed amplitudes (150 and 250 AU) for varying frequencies and (**B**) Micropump pressure output at fixed frequencies (560 and 800 Hz) for varying amplitudes. The maximum pressure output was achieved at an amplitude of 250 and between 525 and 575 Hz.

**Figure 3 micromachines-13-01823-f003:**
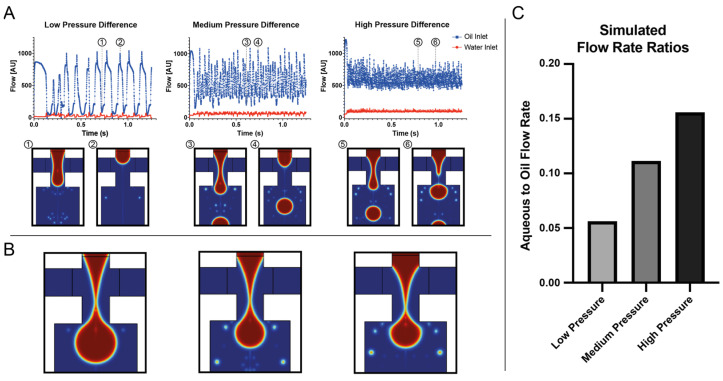
Simulation of microfluidic droplet generation behavior with different applied negative pressures at the outlet. (**A**) shows flows through the respective inlets. The respective flows are oscillating through the droplet break-off, indicating an increase in droplet rate the higher the applied pressure differential. Images (1) and (3) and (5) depict a snapshot of the nozzle at minimal oil flow, (2) and (4) and (6) show the nozzle at maximal oil flow. (**B**) shows the respective droplets immediately prior to break-off. (**C**) shows the dispersed aqueous phase flow normalized to the continuous oil phase flow for the three simulated pressure differentials, indicating an increase in the aqueous-to-oil ratio for increasing negative pressures.

**Figure 4 micromachines-13-01823-f004:**
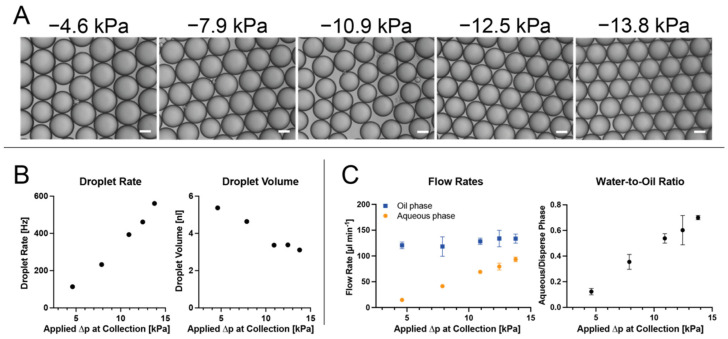
Single−Pump Droplet Generation. (**A**) shows representative images of the resulting emulsions at varying applied pressures at the collection outlet; the white scale bar denotes 100 µm. (**B**) shows droplet rate and volume, the flow rates for both phases, and the water−to−oil ratio for the tested pressures. Notably, the droplet rate increases faster than the droplet volume decreases. (**C**) depicts the flow rate measurements examined by weighing the output after 2 min of operation. Evidently, an increase in the applied negative pressure at the outlet predominantly increases the aqueous phase flow, resulting in an increase in the water−to−oil ratio.

**Figure 5 micromachines-13-01823-f005:**
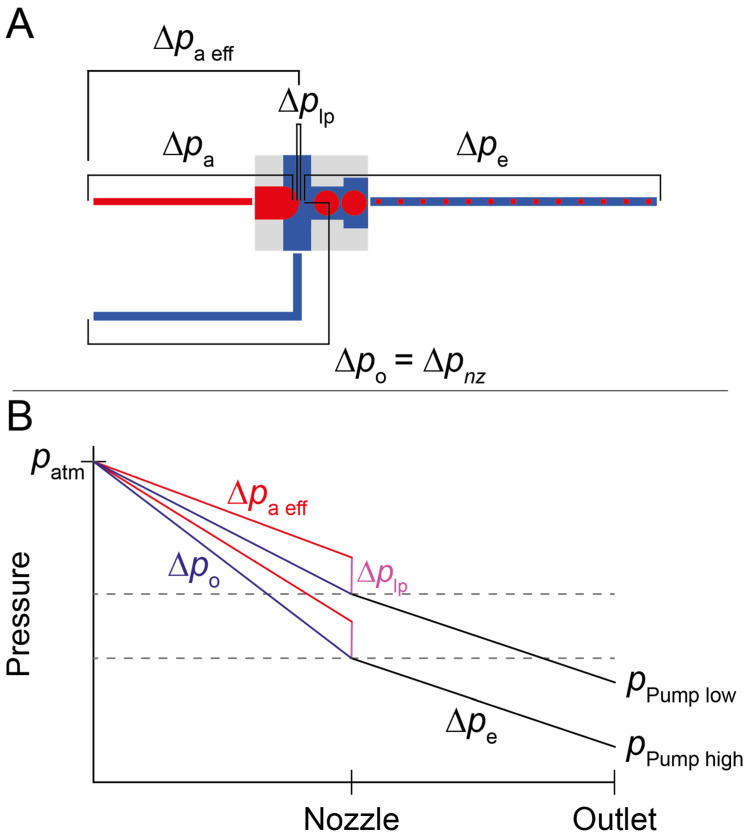
Pressures in the microfluidic system. (**A**) shows a schematic depiction of the microfluidic circuit with the respective pressure differentials and with the two oil channels collapsed into one. Red depicts the aqueous phase, blue the continuous phase. (**B**) represents the idealized pressure along the circuit geometry. Note that ∆*p_lp_* is not constant in reality, however, the change is minuscule enough to be disregarded for our purposes.

**Figure 6 micromachines-13-01823-f006:**
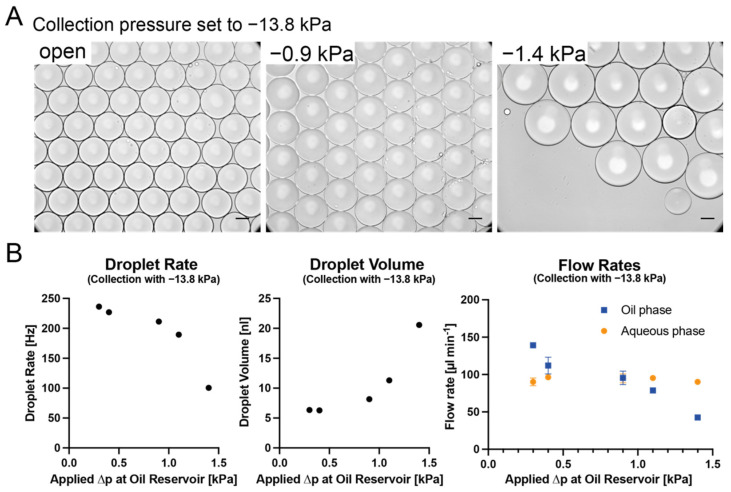
Dual-pump configuration droplet generation. (**A**) shows representative images of the emulsion resulting from −13.8 kPa applied at the outlet, with varying pressures applied to the oil reservoir. Black scale bar denotes 100 µm. (**B**) shows droplet rate and volume and the flow rates for both phases with different pressures applied at the oil reservoir. An increase in the applied negative pressure at the oil reservoir decreases the droplet rate while increasing the droplet volume. The aqueous flow rate is largely unaffected by this, whereas the oil flow rate decreases.

**Figure 7 micromachines-13-01823-f007:**
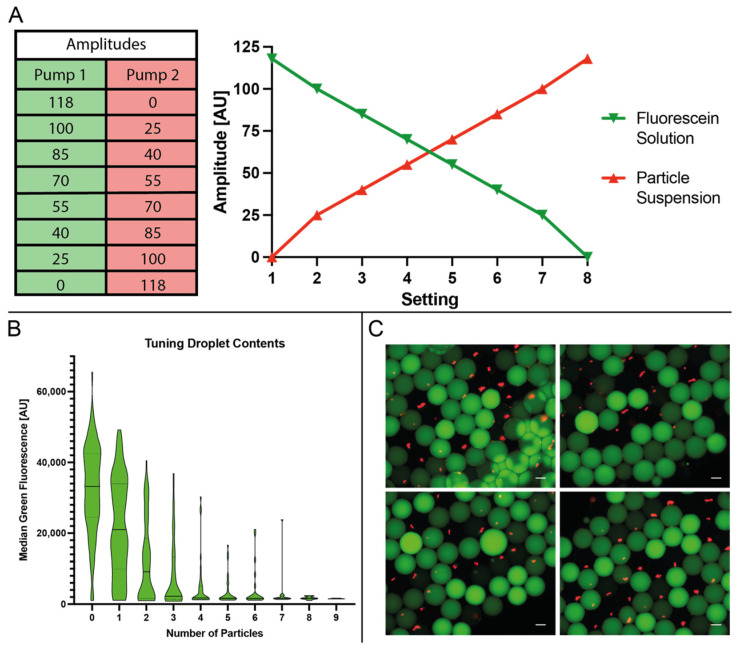
Composition tuning using two pumps at the aqueous inlets. (**A**) illustrates the various settings for dynamic mixing of the two different aqueous samples. (**B**) is a plot of individual droplets’ median green fluorescence over their respective particle content. (**C**) shows four representative images of the emulsion used to obtain the results, combining both fluorescent channels in a composite image. White scale bar represents 100 µm.

**Figure 8 micromachines-13-01823-f008:**
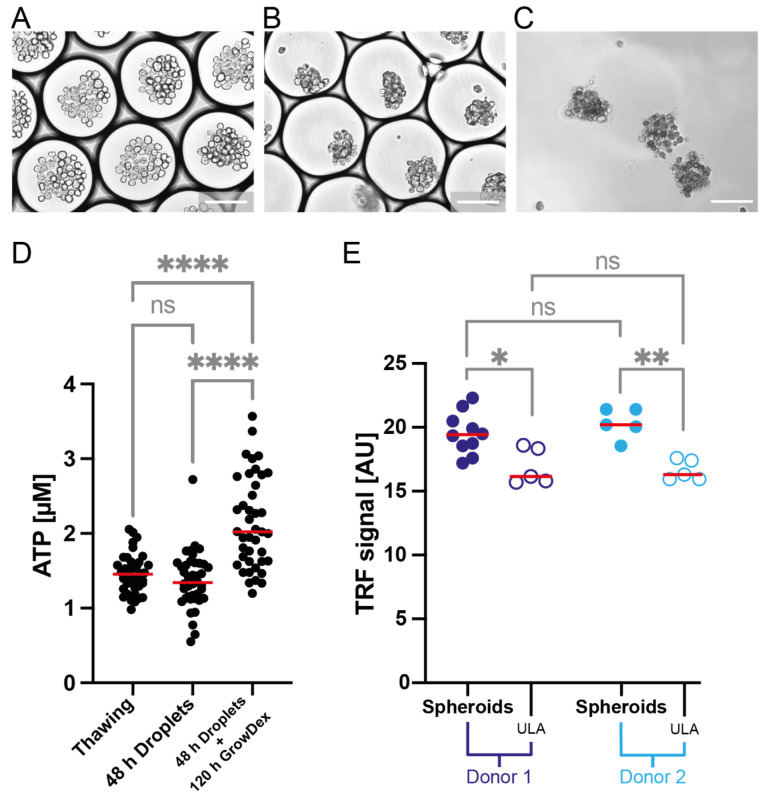
The microfluidic droplet format supports functional cell spheroid formation from primary cells. (**A**–**C**) show representative brightfield images of primary cell spheroid formation using microfluidic droplets generated with the minimal droplet generation setup. White scale bars indicate 100 µm. (**A**) shows the emulsion directly after encapsulation of primary hepatocytes. (**B**) is an image of the emulsion after 24 h of incubation in a syringe, at 37 °C, 5% CO^2^, and 100% humidity. (**C**) shows the spheroids after breaking of the emulsion after 24 h of incubation, demonstrating cohesiveness of the cell assemblies. (**D**) Plot showing ATP concentration as a proxy for viability. Viability of the cells is maintained between thawing and 48-h droplet incubation and breaking of the emulsion. After another 120-h incubation in GrowDex, the viability, as measured by ATP concentration, increases, indicating proliferation. (**E**) compares the functionality of primary hepatocyte spheroids formed in microfluidic droplets encapsulated with the minimal droplet generation setup and spheroids formed in ULA plates from two different sample donors using an albumin quantification assay. Both samples were assessed after 7 days (droplet spheroids were inicubated for 48 h in droplets + 120 h in GrowDex; ULA sample were inicubated for 168 h in ULA plates). In both examined cases, albumin secretion appears to be higher in spheroids assembled in microfluidic droplets. The four stars indicate a *p*-value of <0.001, two stars a *p*-value of <0.01, one star a *p*-value of <0.05, ns indicates insignificance (analyzed with a one-way ANOVA).

## Data Availability

Not applicable.

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
