# Peer review of "A Portable, Negative-Pressure Actuated, Dynamically Tunable Microfluidic Droplet Generator"

_micromachines, 2022, doi:10.3390/mi13111823_

Round 1
Reviewer 1 Report
This work developed a low-cost, minimal footprint droplet generation setup using one to three small membrane pumps. The modular setup is capable of producing monodisperse droplets for a wide range of droplet sizes and can be dynamically adjusted to vary droplet composition. It is an interesting work, I suggest a major revision in order to better benefit the general audience of Micromachines.
1. In Figure 4, the diameter of droplet at P = -10.9 kPa is similar to P = -12.5 kPa. When P = -13.8 kPa, why the diameter of droplet deceases?
2. Can the droplet generator produce smaller size droplet (diameter<100 μm)?
3. In Figure 6A, the droplet is not uniform at P = -1.4 kPa.
4. How about the stability of the droplet?
5. In Figure 8, it shows cell spheroid formation. Can it achieve single cell in a droplet?
Reviewer 2 Report
1. The research contents was not well organized and the explanation was not clear. Especially simple explanations were repeated in 2. Materials and Methods; The description of the experiment was duplicated.
2. Also, the title and contents of the paper do not match well. The droplet generator system proposed in this paper is said to be portable. However, in this regard, not only the explanation is insufficient, but also the comparative explanation with the existing system is lacking. What is the overall system size? If it is portable, it is very important thing.
3. For the system proposed in this paper, what is improved and what are the advantages compared with the general negative-pressure generator should be explained in detail.
4. There is no description of the chip design used, only the description of the entire setups. If the general design of flow focusing droplet generator is used, the simulation is not new at all because it is not different from the existing one. Please explain about what’s different from other previous researches in Fig. 3?
5. In the case of a flow focusing droplet generator, the droplet size changes depending on the pressure change. It is necessary to explain how the system proposed in this paper maintains the pressure constant and how much the droplet size fluctuates according to the minute pressure change in Fig. 5.
Round 2
Reviewer 1 Report
I would like to recommend the acceptance of this manuscript for publishing.